# Mode Switching in a Compressible Rectangular Cavity Flow

Yi-Xuan Huang [1] and Kung-Ming Chung [2,*]

1    Department of Aeronautics and Astronautics, National Cheng Kung University, Tainan 701, Taiwan;
     qwer22147@gmail.com
2    Aerospace Science and Technology Research Center, National Cheng Kung University, Tainan 711, Taiwan
*    Correspondence: kmchung@mail.ncku.edu.tw; Tel.: +886-6-239281

**Abstract:** This study determines the mean and fluctuating pressures for flow through a rectangular shallow cavity (ratio between the length and the depth = 2.43, 4.43, and 6.14; ratio between the length and the width = 0.5, 1.0, and 2.0) at a Mach number of 0.64 in a blowdown transonic wind tunnel. A amplitude modulation analysis is used for the post-processing of the fluctuating pressure signals. The spectral analysis (wavelet) shows the intermittent behavior of the discrete Rossiter–Heller modes. A correlation analysis determines that mode switching is more significant between the second and third modes (organized structures that are associated with shear layer vortices), particularly for two-dimensional and shallower cavities.

**Keywords:** shallow cavity; resonant frequency; mode switching





## 1. Introduction

A cavity flow is characterized by the distribution of the longitudinal surface pressure on the cavity floor [1–3]. The type of flowfield (open, transitional, or closed) is primarily a function of the ratio between the length, *l*, and the depth, *h* (*l*/*h*). For an open cavity flow (*l*/*h* < 6–8), an unsteady shear layer spans the cavity and impinges near the trailing edge [4,5]. There is a uniform longitudinal pressure distribution on the cavity floor. Two distinct separation regions form downstream of the front face and upstream of the rear face in a closed cavity flow (*l*/*h* > 10–15). The ratio between *l* and the width, *w*, (*l*/*w*) also characterizes a cavity flowfield [6]. The degree of three-dimensionality is associated with the value of *l*/*w*, which results in a spanwise oscillation for allocation at which there is shear layer impingement [7].

Landing gear wells or weapon bays are classified as open cavities. The incoming boundary layer separates at the leading edge, and the shear layer rolls up into vortices. The flow is subject to global instabilities (Kelvin–Helmholtz instabilities) because there is a difference in the velocity of the freestream and the velocity of the fluid inside a cavity [8]. Large-scale vortical structures are subject to convection downstream and attach near the trailing edge. Acoustic waves then propagate upstream toward the cavity's front face and disturb the upstream shear layer to create another vortex. This is known as a feedback loop (downstream convection instabilities and upstream propagating acoustic waves), which leads to self-sustained oscillations and discrete tones [9]. The pressure fluctuations for an open cavity flow are considerably greater than those for a closed cavity flow. There is intrinsic unsteadiness in the recirculation region because of the instabilities in the shear layer and oscillations that originate in the transverse dimension of the cavity [10].

Rossiter [11] proposed a semi-empirical formula to calculate non-harmonic modal frequencies. Heller and Bliss [12] determined that the temperature in an open cavity flow affects the speed of sound. The empirical Rossiter–Heller (R–H) equation (Strouhal number, $St_n$) is written as Equation (1).

$$St_n = \frac{f_n l}{U_\infty} = \frac{n - \alpha}{M\left(1 + \frac{\gamma - 1}{2}M^2\right)^{-\frac{1}{2}} + \frac{1}{k_c}} \tag{1}$$

where $n$ is the number of discrete modes. $f_n$ and $U_\infty$ are the discrete modal frequency and the freestrem velocity, respectively; the empirical parameters ($k_c$ and $\alpha$) vary with cavity geometry and freestream Mach number, $M$ [13], $k_c$ (=0.50–0.75) is the ratio of the perturbed convective velocity and $U_\infty$. $\alpha$ (=0.25 or 0.062 $l/h$) is the phase lag (expressed as a fraction of wavelength), which corresponds to the lag time between the passage of a vortex and the emission of an acoustic pulse [14].

The spectral characteristics of resonance in an open cavity flow are defined using a short-time Fourier transform [15–17]. The first R–H mode ($R_1$) corresponds to large-scale motions in the shear layer and in the vicinity of the recirculation region. The second and third R–H modes ($R_2$ and $R_3$) feature organized structures that are associated with shear layer vortices [18]. However, the R–H modes do not necessarily occur simultaneously, and the dominant energy can vary with time (a process for temporally dominant energy shifts or mode switching).

Kegerisea et al. ($M$ = 0.04–0.8) [19] used a wavelet transform to determine mode switching. The R–H modes do not exhibit strong nonlinear coupling, as verified by the study by Pandian et al. ($M$ = 1.8) [20]. Gloerfelt et al. ($M$ = 0.4–0.8) [21] determined that there is switching between $R_1$ and $R_2$ for cavity oscillations. Bacci et al. ($M$ = 0.81) [10] showed that the energy distribution is rearranged between the R–H modes. Observations of the velocity and pressure fields showed that the ability to detect mode switching is dependent on the location within the cavity and the R–H mode number [18].

Mode switching is an evolving research field for the dynamics of flow in an open cavity. However, there is very limited data on this unstable phenomenon. This study uses time-frequency analysis for the post-processing of fluctuating pressure signals that are recorded on the floor of rectangular cavities at $M$ = 0.64. The effect of $l/h$ and $l/w$ (degree of three-dimensionality) on the mode amplitude over time is determined, as is the intermittent nature of the R–H modes. Mode switching is quantified using an amplitude demodulation technique and correlation coefficients. Before discussing the results, details of the experiment setup are outlined next.

## 2. Experimental Setup

### 2.1. Transonic Wind Tunnel

The experiments were conducted in a blowdown-type wind tunnel at the Aerospace Science and Technology Research Center, National Cheng Kung University (ASTRC/NCKU). The facility has two compressors, two air dryers, three storage tanks (volume = 180 m$^3$) that allow a maximum pressure of 5.15 MPa, a hydraulic system, and a tunnel (a stilling chamber, a nozzle, and a test section). The test section within a plenum chamber is 600 mm square and 1500 mm in length. Acoustic waves in the test section are mitigated by using solid sidewalls, perforated top and bottom walls (inclined holes with 6% porosity). A rotary perforated sleeve valve controls the stagnation pressure, $p_o$, which ranges from 137.8 to 344.5 kPa. The stagnation temperature is room temperature. The operational Mach number is 0.2–1.4, and the maximum Reynolds number is $3.5 \times 10^7$ per meter. The test conditions are monitored and recorded using a National Instruments system (NI PXIe-8840 RT, PXI-7846, PXI-6511, and PXI-6513; Austin, TX, USA).

### 2.2. Test Model

The test model was a 450-mm-long flat plate with a 4° sharp leading edge and a 180-mm instrumentation plate with an open cavity. Pressure taps were machined along the centerline of the rectangular cavities (upstream/downstream locations and cavity floor), as shown in Figure 1. The model was supported by a single foot that was fixed to the bottom wall of the test section. A turbulent boundary layer developed naturally upstream of an

open cavity. The normalized boundary layer profile using a Pitot probe and a transversing device was in the 1/7 velocity power law [22].

Six cavity models were fabricated. The geometry for each cavity is shown in Table 1. For cavities 1–3, the value of $h$ (=7 mm) was fixed and the value of $l$ (=17–43 mm) was varied, which corresponds to a specific value for $l/h$ of 2.43, 4.43, and 6.14 (open-type cavities). The value of $l/w$ is an indication of the degree of three-dimensionality of a cavity flow [6]. The value of $w$ was varied for the same value for $l$ (=43 mm) and $h$ (=9.7 mm) for cavities 4–6 ($l/h$ = 4.43). The values for $l/w$ were 0.5, 1.0, and 2.0. Side fences on both sides of the instrumentation plate were used to prevent crossflow.

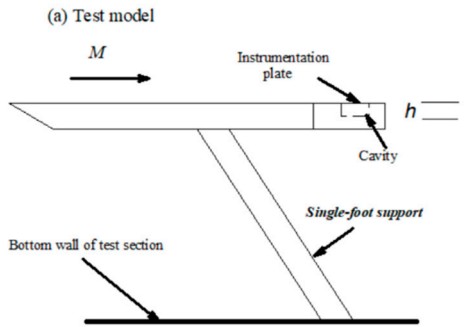

**Figure 1.** Test configuration.

**Table 1.** Cavity dimensions.

| Cavity | $l$ | $h$ | $w$ | $l/h$ | $l/w$ |
|---|---|---|---|---|---|
| 1 | 17 | 7 | 17 | 2.43 | 1 |
| 2 | 31 | 7 | 31 | 4.43 | 1 |
| 3 | 43 | 7 | 43 | 6.14 | 1 |
| 4 | 43 | 9.7 | 86 | 4.43 | 0.5 |
| 5 | 43 | 9.7 | 43 | 4.43 | 1 |
| 6 | 43 | 9.7 | 21.5 | 4.43 | 2 |

*2.3. Instrumentation and Data Acquisition System*

The value of $M$ was 0.64 ± 0.01. The stagnation pressure, $p_0$, was 172 ± 0.5 kPa, and the stagnation temperature was room temperature. The boundary layer thickness, $\delta$, at 475 mm from the leading edge was 7.3 ± 0.2 mm [23], and the Reynolds number, $Re_\delta$, was $1.63 \times 10^5$.

Mean and fluctuating surface pressures were measured using flush-mounted Kulite pressure transducers (Model XCS-093-25A, B screen; Leonia, NJ, USA), which were powered by a DC power supply (GW Instek PSS-3203; Taipei, Taiwan) of 10.0 V. The nominal outer diameter of the Kulite sensors is 2.36 mm, and the sensing element is 0.97 mm in diameter. Crocos [24] showed that the sensitive area of a pressure transducer limits the temporal resolution of pressure signals. For a convection velocity of $0.8\, U_\infty$, the maximum measurable frequency for this study is approximately 63 kHz. External amplifiers (Ectron Model 753A; San Diego, CA, USA) with a gain of 20 were used to increase the signal-to-noise ratio. A National Instruments device (NI PXI-6123; Austin, TX, USA) triggered all input channels and recorded data. The sampling rate was 5 μs (200 ksamples/s).

*2.4. Mean and Fluctuating Pressure Coefficients*

Each measurement consisted of 131,072 data points that were divided into 32 subsets. The mean and fluctuating pressures are determined using Equations (2) and (3):

$$\bar{p}, p_w = \frac{1}{N}\sum_{j=1}^{N} p(t_j) \tag{2}$$

$$\sigma_p = \sqrt{\frac{1}{N-1}\sum_{j=1}^{N}\left[p(t_j) - \bar{p}\right]^2} \tag{3}$$

where $p(t_j)$ is the instantaneous pressure signal, and $N$ is the number of data points.

The uncertainty is determined for flow through a flat plate. The respective values for the mean, $C_p$ ($=(p_w - p_\infty)/q$), and fluctuating pressure, $C_{\sigma_p}(=(\sigma_p - \sigma_{p,\infty})/q)$, coefficients are 0.43% and 0.13% [23], $q$ is the dynamic pressure, and the subscript $\infty$ denotes the freestream value.

*2.5. Spectral Analysis*

For a set of pressure signals, $p_n(t_j)$, that are sampled using a time step of $\Delta t$, the discrete Fourier transform is defined as:

$$p_n(f_k) = \sum_{j=1}^{N} p_n(t_j)\cdot e^{-\frac{2\pi kji}{N}} \tag{4}$$

where $n = 1, 2, 3, \ldots, 32$, $j = 1, 2, 3, \ldots, 4096$ $(N)$

The one-side power spectral density (PSD) is:

$$PSD_n(f_k) = \frac{2}{f_s\cdot N}\cdot p_n(f_k)\cdot p_n^*(f_k), k = 1, 2\ldots, \frac{N}{2} + 1 \tag{5}$$

where $f_k = \left\langle 0, 1, 2, \ldots \frac{N}{2} \right\rangle \cdot \frac{f_s}{2N}$

The PSD analysis uses 31 subsets with a 50% overlap, and a Hanning window is used to reduce leakage effects. The frequency resolution is 48.8 Hz. The ensemble PSD is determined by averaging the values for all subsets.

Wavelet analysis is a generalization of the time-frequency transform and is used to identify dynamic structures in both time and space (intermittent and non-stationary signals) [25]. It projects a signal into a set of base functions, which are wavelets. A continuous wavelet transform is defined as:

$$W(\tau, a) = \int_{-\infty}^{\infty} p(t)\cdot \Psi_{a,\tau}^*(t)dt \tag{6}$$

where $\Psi_{a,\tau}^*(t)$ is the wavelet function and the asterisk represents the complex conjugate. A Morlet mother wavelet is used:

$$\Psi_{a,\tau}^*(t) = a^{-1/2}e^{j\omega_\psi\left(\frac{t-\tau}{a}\right)}\cdot e^{-\left(\frac{t-\tau}{a}\right)^2/2} \tag{7}$$

where $a > 0$ is the scale, and $\tau$ is the time delay. The mother wavelet is normalized by $a^{-1/2}$ to ensure that all sets of the mother wavelet with different scales have the same energy. Torrence and Compo [26] showed that a wavelet transform can be represented by an inverse Fourier transform, as shown in Equation (8):

$$W(t, a) = \int_{-\infty}^{\infty} p(\tau)\cdot a^{-\frac{1}{2}}e^{-\frac{\left(\frac{\tau-t}{a}\right)^2}{2}}\cdot e^{-j\omega_\psi\left(\frac{\tau-t}{a}\right)} \tag{8}$$

The term $a^{-1/2}e^{-\left(\frac{\tau-t}{a}\right)^2/2}$ is similar to a moving Gaussian window function. The local scale, $a$, determines the peak amplitude and the decay rate, which are correlated with frequencies. Equation (9) shows that small scales have a higher resolution in terms of time but a lower resolution in terms of because $-\Delta f/f$ is a constant (a constant percentage bandpass filter):

$$\frac{1}{f} = \frac{4\pi}{\omega + \sqrt{2 + \omega^2}}\frac{1}{a} \tag{9}$$

*2.6. Amplitude Demodulation Technique*

The joint time-frequency analysis shows that interactions can occur between the R–H modes in an open cavity flow [10,18,21], but it is difficult to quantify mode switching. However, if there is energy transfer between the R–H modes, mode switching can be quantified using amplitude demodulations. These modulations result in a transfer of energy to the sidebands of spectral peaks and contribute to a broadening of the spectrum [27].

According to Equation (1), the surface pressure fluctuating signal $p'(t)$ is a combination of the amplitude of the R–H modes and the broadband noise, $z(t)$, as follows:

$$p'(t) = \sum_1^m A_m \cdot \cos\left(2\pi f_m t + \rho_m\right) + z(t), \ m = 1, 2, \ldots \tag{10}$$

where $A_m$ and $\rho_m$ are the respective amplitude and the phase as a function of time. Multiplication by a complex exponential at the modal frequency of interest, $f_k$,

$$p'(t) \cdot e^{-i(2\pi t f_k)} = \sum \frac{A_m}{2} \left[ e^{i(2\pi t (f_m - f_k) + \rho_m)} - e^{-i(2\pi t (f_m + f_k) + \rho_m)} \right] + z(t) \cdot e^{-i(2\pi t f_k)} \tag{11}$$

Each component at a given value for $f_k$ is shifted down to frequency $(f_m - f_k)$. A low-pass filter (LPF) is then used to separate the frequencies.

Equation (12) defines the relationship between the first three R–H modes [11], which is

$$f_2 \approx 2.33 f_1, f_3 \approx 3.66 f_1 (M = 1.5 \sim 0.3) \tag{12}$$

The cut-off frequency for a LPF is used to isolate the energy that is near the frequency of interest. An example of the amplitude demodulation for $l/h = 6.14$ is shown in Figure 2. The red lines represent fluctuations for the first three R–H modes, and the blue lines correspond to the variation in amplitude with time. There is an increase in the modal frequency for a higher mode. For upstream propagating acoustic waves, the wave length, $\lambda_i$, is determined as $a/f_i$, where $a$ is the speed of sound. The value of $l$ is approximately $2\lambda_1$, $4\lambda_2$, or $6.7\lambda_3$. The coupling between R–H modes and acoustic waves would result in greater amplitude for the first and second modes. The standard deviation of the R–H modes, $C_{A,Rossiter}$, is determined using the amplitude, $A_j$, for the first three R–H modes (blue lines), as shown in Equation (13).

$$\sigma_{Rossiter} = \sqrt{\frac{1}{N-1} \sum_{j=1}^N \left[ LPF\left\{ p'(t_j) \right\} \right]^2} = \sqrt{\frac{1}{N-1} \sum_{j=1}^N [A_j]^2}$$

$$C_{A,Rossiter} = \left( \sum_{i=1}^3 \sigma_{Rossiter,i} \right) / q \tag{13}$$

Spearman's correlation coefficient , $\gamma_{ij}$, (Equation (14)) [28] is used to quantify mode switching between the R–H modes (blue lines in Figure 2). If the correlation coefficient is negative, there is a decreased monotonic trend between the R–H modes.

$$\gamma_{ij} = \frac{cov\left( R(A_i), R(A_j) \right)}{\sigma_{R(A_i)} \sigma_{R(A_j)}} i, j = 1, 2, \ldots; i \neq j \tag{14}$$

where $cov\left( R(A_i), R(A_j) \right)$ is the covariance of rank variables and $\sigma_{R(A_i)}$ and $\sigma_{R(A_j)}$ are the standard deviation for rank variables.

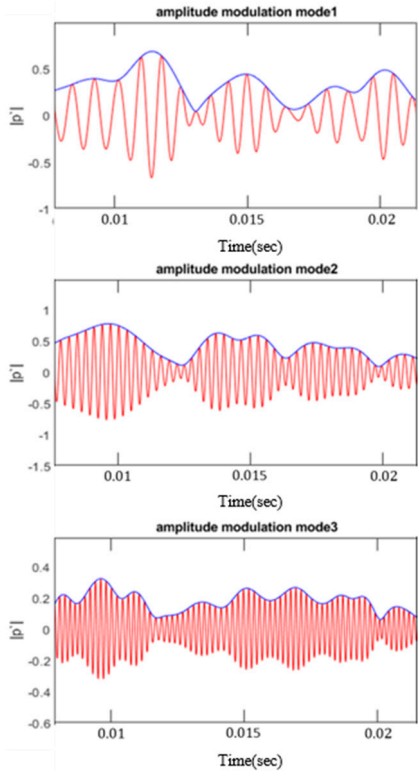

**Figure 2.** Amplitude demodulation for the R–H modes for $l/h$ = 6.14.

## 3. Results and Discussion

### 3.1. Mean and Fluctuating Pressure

The distribution of $C_p$ along the centerline for cavities 1–3 ($l/w$ = 1.0) is shown in Figure 3a. The horizontal axis is the normalized streamwise location, $x^*$ (=$x/l$), and the origin is at the front face on the cavity floor. There is a decrease in the value of $C_p$ downstream of the front face, particularly for $l/h$ = 2.43, followed by an increase further downstream because the shear layer diffuses (expansion) and is deflected (compression) [29]. The value of $C_p$ decreases near the rear face, and there is a recovery process downstream. The effect of $l/w$ on the $C_p$ distribution is shown in Figure 3b. For $l/h$ = 4.43 (cavities 4–6), there is a similar effect on the distribution of $C_p$ as $l/w$ varies. For $l/w$ = 2.0, an increase in the degree of three-dimensionality affects the $C_p$ distribution. There is an increase in the value of $C_p$ upstream of the front face and a decrease on the cavity floor. The pressure gradient is reduced near the rear face because the spanwise velocity is greater [30].

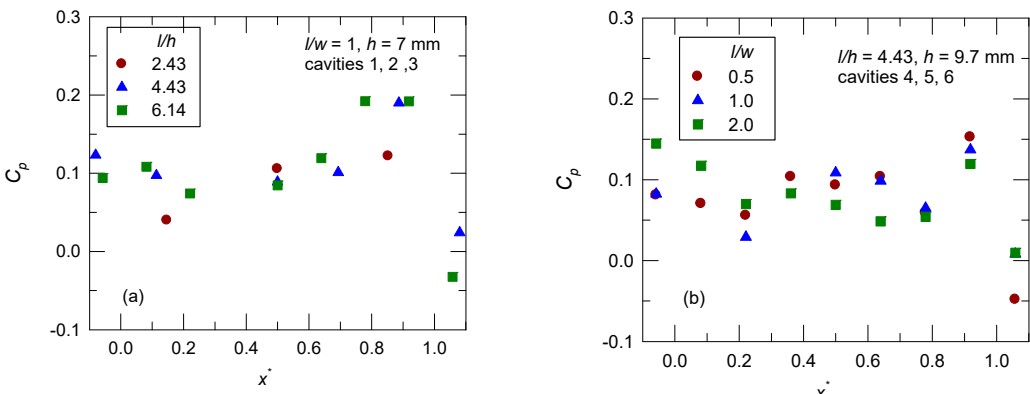

**Figure 3.** Streamwise mean pressure distribution: (**a**) $l/w$ = 1.0; (**b**) $l/h$ = 4.43.

Figure 4a shows the distribution of $C_{\sigma_p}$ for $l/w$ = 1.0 (cavities 1–3, $h$ = 7 mm). There is a flatter distribution of $C_{\sigma_p}$ for $l/h$ = 2.43. For $l/h$ = 4.43 and 6.14, expansion over the leading edge causes damping downstream of the front face (a lower value in $C_{\sigma_p}$). Self-sustained oscillations (downstream convection instabilities and upstream propagating acoustic waves) result in a peak value of $C_{\sigma_p}$, $C_{\sigma_p,max}$, near the rear face, which corresponds to the shear layer impingement and the process for mass addition/removal. It is also noted that there is a decrease in the value of $C_{\sigma_p}$ for $l/h$ = 4.43 downstream of the rear face.

The effect of $l/w$ (cavities 4–6, $l/h$ = 4.43, $h$ = 9.7 mm) is shown in Figure 4b. The distribution of $C_{\sigma_p}$ is approximately the same as that for $l/w$ = 0.5 and 2.0. The value of $C_{\sigma_p}$ is less on the cavity floor for $l/w$ = 1.0. An increase in the three-dimensionality ($l/w$ = 0.5) results in a greater value in $C_{\sigma_p}$. For cavities 2 and 5 ($l/h$ = 4.43; $l/w$ = 1.0), an increase in the value of $h$ results in greater $C_{\sigma_p}$ downstream of the rear face. This implies the effect of $h/\delta$ on the shear layer impingement and the process for mass addition/removal.

The effect of $l/h$ on $C_{\sigma_p,max}$ for $l/w$ = 1.0 (cavities 1–3, $h$ = 7 mm) is shown in Figure 5a. An increase in the value of $l/h$ induces stronger self-sustained oscillation, so the value of $C_{A,Rossiter}$ increases. This shows that the first three R–H modes dominate self-sustained oscillations for an open cavity flow. For $l/h$ = 4.43, variation in the value of $l/w$ (cavities 4–6, $h$ = 9.7 mm) has a minor effect on the amplitude of $C_{\sigma_p,max}$, as shown in Figure 5b. There is a decrease in the value of $C_{A,Rossiter}$ as $l/w$ increases. This indicates that variation in $C_{\sigma_p,max}$ depends on the value of $l/h$ and the first three R–H modes are more dominant as the value of $l/w$ decreases.

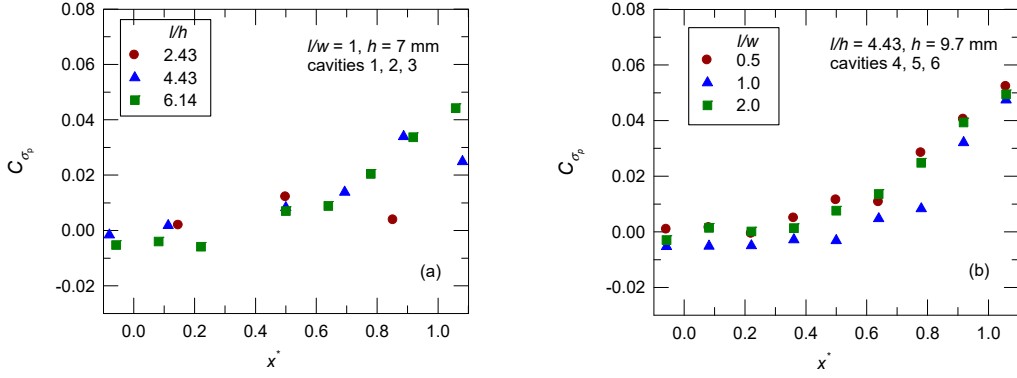

**Figure 4.** Streamwise fluctuating pressure distribution: (**a**) $l/w$ = 1.0; (**b**) $l/h$ = 4.43.

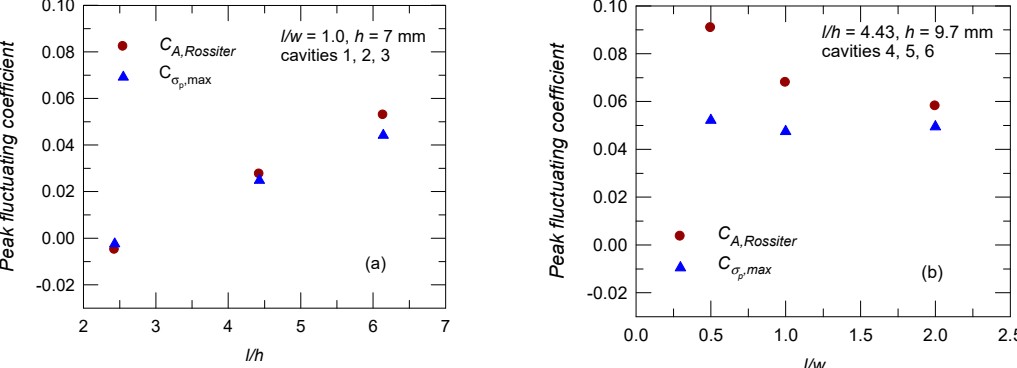

**Figure 5.** Peak fluctuating pressure: (**a**) $l/w$ = 1.0; (**b**) $l/h$ = 4.43.

### 3.2. Spectral Analysis

The frequency of the R–H modes, which is calculated using Equation (1), is shown in Table 2a. There is a decrease in the value of the first three R–H modes ($f_1$, $f_2$, and $f_3$) as $l$ increases. The modal frequencies are also determined using fluctuating pressure signals, and the results are shown in Table 2b. This experimental data is in reasonable agreement

with the R–H modes for cavities 1–3 ($l/w$ = 1.0, $h$ = 7 mm), and the discrepancy is less than 10%, as are $f_2$ and $f_3$ for cavity 5 ($l/w$ = 1.0, $h$ = 9.7 mm). For $l/h$ = 4.43 (cavities 4–6), the value of $f_1$–$f_3$ is approximately the same for all three test cases. The discrepancy between the experimental data and R–H modes is significant (up to 30%), particularly for $l/w$ = 0.5, which corresponds to the effect of the degree of three-dimensionality.

Figure 6 shows a plot of the PSD near the rear face for $l/h$ = 6.14 and $l/w$ = 1.0. The frequency resolution for the PSD is 48.8 Hz. There are multiple peaks at $f_1$ = 1495 Hz, $f_2$ = 3326 Hz, and $f_3$ = 5310 Hz, which correspond to the first three R–H modes. The second mode is more dominant. Kegerise et al. [19] showed that nonlinear quadratic interaction can occur between the R–H modes, and other peaks (the sums and differences of $f_1$, $f_2$, and $f_3$) are detected. However, this study does not demonstrate this phenomenon.

The PSD does not show that modes coexist or that mode switching occurs. The spectrum for $l/h$ = 6.14 that is calculated using a wavelet analysis is shown in Figure 7. The time range in the plot is from 0 to 0.25 s, and the frequency range is chosen in order to cover the first three R–H modes. The amplitude varies over time, and there is intermittent behavior, which is not shown in Fourier spectra. The figure also shows that there is no evidence of a monotonic relationship between modes.

**Table 2.** Frequency of Rossiter–Heller modes: (**a**) prediction; (**b**) experiment.

| (a) | | | |
|---|---|---|---|
| **Cavity** | $f_1$, **Hz** | $f_2$, **Hz** | $f_3$, **Hz** |
| 1 | 3864 | 8983 | 14,116 |
| 2, 4–6 | 2119 | 4926 | 7741 |
| 3 | 1528 | 3551 | 5580 |

| (b) | | | |
|---|---|---|---|
| **Cavity** | $f_1$, **Hz** | $f_2$, **Hz** | $f_3$, **Hz** |
| 1 | 4339 | 8679 | 12,902 |
| 2 | 2319 | 4639 | 7446 |
| 3 | 1495 | 3326 | 5310 |
| 4 | 1635 | 3540 | 5480 |
| 5 | 1513 | 3446 | 5419 |
| 6 | 1757 | 3564 | 5297 |

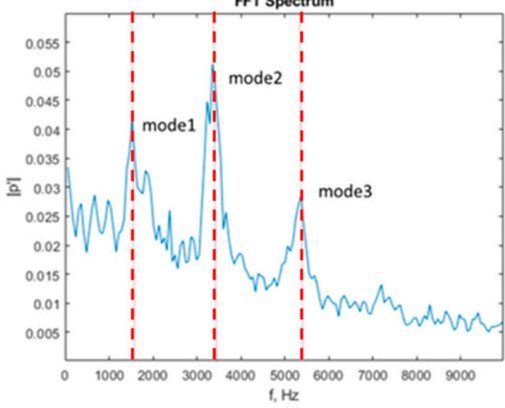

**Figure 6.** PSD for $l/h$ = 6.14.

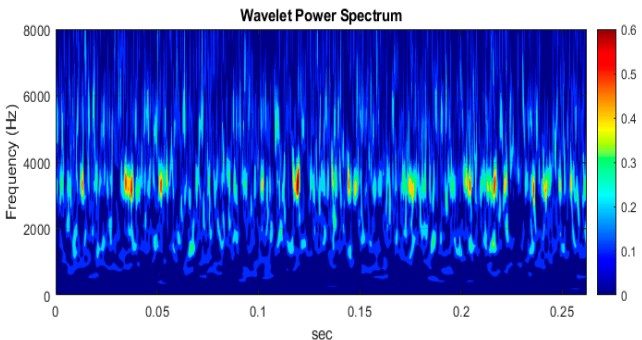

**Figure 7.** Wavelet transform for $l/h$ = 6.14.

Previous studies [19,20] showed that the R–H modes do not exhibit strong nonlinear coupling. This study uses the correlation between the R–H modes, which refers to the variation of amplitude over time that is shown in Figure 2, to determine the degree of mode switching. For $l/w$ = 1.0 (cavities 1–3), as shown in Figure 8a, the value of $\gamma$ (= −0.15 to 0.09) is approximately the same for $l/h$ = 4.43 and 6.14. Since $R_2$ and $R_3$ feature organized structures that are associated with shear layer vortices [18], a negative value for $R_2R_3$ denotes a decreased monotonic trend or competition for available energy for these two modes. For $l/h$ = 2.43 ($\gamma$ = −0.12 to 0.02), mode switching is less significant than that for shallower cavities ($l/h$ = 4.43 and 6.14).

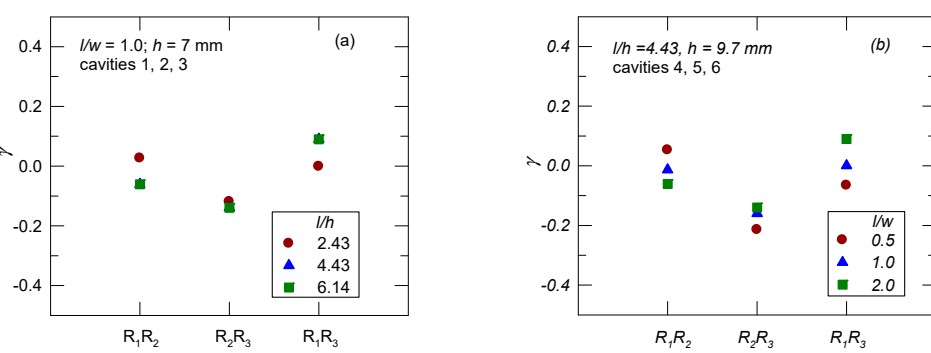

**Figure 8.** Correlation between the R–H modes: (**a**) $l/w$ = 1.0; (**b**) $l/h$ = 4.43.

For $l/h$ = 4.43 (cavities 4–6), the effect of $l/w$ is shown in Figure 8b. The value of $\gamma$ for $R_1R_2$ and $R_1R_3$ is −0.06–0.05, which shows that there is less mode switching. There is a slight increase in the value of $\gamma$ as $l/w$ increases, possibly because of spanwise oscillation at the location where shear layer impingement occurs for a three-dimensional rectangular cavity flow [7]. The value of $l/w$ has an opposite effect on $R_1R_2$ and $R_1R_3$.

## 4. Conclusions

Self-sustained oscillations in a flow through an open cavity are associated with multiple acoustic tones with discrete frequencies (Rossiter–Heller modes), which depend on the geometry of the cavity. A wavelet analysis is used to analyze fluctuating pressure signals with discrete tones (of an intermittent nature). This study determines the effect of $l/h$ and $l/w$ on the strength of variations as a function of both frequency and time for six cavity models (cavities 1–3: $l/w$ = 1.0, $l/h$ = 2.43, 4.43, and 6.14; cavities 4–6: $l/h$ = 4.43, $l/w$ = 0.5, 1.0, and 2.0) at $M$ = 0.64. Amplitude demodulation is used to trace the resonance and better describe the self-sustained oscillations. The correlation between the Rossiter–Heller modes shows that mode switching is more significant for two-dimensional and shallower cavities, particularly between the second and third modes. This implies that the energy distribution is rearranged for feature-organized structures that are associated with shear layer vortices.

**Author Contributions:** Conceptualization, K.-M.C.; methodology, Y.-X.H.; formal analysis, Y.-X.H.; data curation, K.-M.C. and Y.-X.H.; writing—original draft preparation, Y.-X.H.; writing—review and editing, K.-M.C.; funding acquisition, K.-M.C. All authors have read and agreed to the published version of the manuscript.

**Funding:** This research was funded by the National Science and Technology Council, Taiwan, grant number MOST 110-2221-E-006-097.

**Data Availability Statement:** Data is available upon request.

**Acknowledgments:** The authors are grateful for the technical support of the ASTRC/NCKU technical staff.

**Conflicts of Interest:** The authors declare no conflict of interest.

## Nomenclature

| | |
|---|---|
| $C_p$ | mean surface pressure coefficient |
| $C_{\sigma_p}$ | fluctuating pressure coefficient |
| $C_{\sigma_p,max}$ | peak fluctuating pressure coefficient |
| $f_i, f_2, f_3$ | frequency of Rossiter–Heller (R–H) modes |
| $h$ | cavity depth |
| $l$ | cavity length |
| $M$ | freestream Mach number |
| $\bar{p}, p_w$ | mean surface pressure |
| $p_o$ | stagnation pressure |
| $p_\infty$ | freestream static pressure |
| $p(t_j)$ | pressure signal |
| $q$ | dynamic pressure |
| $R_1, R_2, R_3$ | Rossiter–Heller modes |
| $St$ | Strouhal number |
| $U_\infty$ | freestream velocity |
| $w$ | cavity width |
| $x$ | coordinate along the centerline of model surface |
| $x^*$ | normalized streamwise distance, $x/\delta$ |
| $\delta$ | incoming boundary-layer thickness |
| $\gamma, \gamma_{ij}$ | Spearman's correlation coefficient |
| $\sigma_p$ | fluctuating (rms) pressure |

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
