# Peer review of "Mode Switching in a Compressible Rectangular Cavity Flow"

_aerospace, doi:10.3390/aerospace10060504_

Round 1
Reviewer 1 Report
The article deals with the problem of the fluctuating pressures for flow through a rectangular shallow cavity. In the paper, six different cavities were investigated and the pressure fluctuation was analyzed. Although the article is done well, the research plan is correct and the analysis is supported by the work of other researchers, I consider the article unnecessary.
The measurement points presented in Fig. 3 to 6 are placed very rarely, and it is very hard to investigate anything on their basis. For this reason, the analysis and discussion are extremely short. Only 4 pages, from 6 to 9 where half of pages 6 and 9, page 7, and above half of page 8 are charts and tables. Spectral analysis is the strength of the article. I suggest adding the error analysis between the prediction and experiment.
The presentation of the problem is unclear. The research stand has been presented very sparingly and without knowing the subject and the specificity of the research undertaken in the article, it is impossible to guess what it looks like. Figure 1 should be refined and dimension h shown in the figure.
The applicability of the research presented should be emphasized in the introduction.
The author Kung-Ming Chung is cited in references 2, 3, 6, 21, 22. It is 5/29 references, which is 17%. Reference 6 is needed in the introduction (let's say), 21 and 22 deal with methodology, but 2 and 3 are boosting citations. Please, change them to other publications.
The references are quite old. Only 5 positions have 5 years, and 12 publications have more than 20 years. Has so little been done on this topic in recent years? Of course, this is not a mistake, but please pay attention to this fact.
Reviewer 2 Report
Comments to authors,
This paper aimed to conduct a comprehensive study on time-averaged and fluctuating pressure measurements through a three-dimensional cavity, experimentally and numerically. Their analysis concludes that mode switching is more significant between the second and third Rossiter-Heller modes.
The topic area is appropriate for Aerospace, and I do believe the manuscript is ready for publication. Reasons for this assessment are provided below in general and specific comments.
I suggest the authors consider the following modifications classified into general and specific comments before resubmitting the paper.
General Comments:
1) I would suggest identifying what new and novel in this study are; in the abstract, the last paragraph in the introduction, and especially in the conclusions sections
2) What is significant about this work and its contribution to the literature? Your revisions to consider this comment would be helpful
3) I recommend adding a few more recently published articles, as most references are old.
For example, Maceda, G.Y.C., Varon, E., Lusseyran, F., Noack, B.R. (2023) Stabilization of a multi-frequency open cavity flow with gradient-enriched machine learning control. Journal of Fluid Mechanics, 955, A20, doi: 10.1017/jfm.2022.1050
4) I would recommend using the same font style in equations and the text. For example, “U” appears differently in equation (1), and the comment follows the equation
5) I would suggest showing the “Test Configuration” figure (Figure 1) in a 3D way to be more informative (Please be advised this is not a must, and ups to the authors to make their decision in this regard)
6) I would suggest merging Figures 3 and 4 to appear as one figure and break it down into (a) and (b)
7) The same comment as 6 for Figures 5 and 6 and also for Figures 7 and 8
8) I would suggest placing the previous three pair figures (comments 6 and 7) side-by-side
Specific Comments:
1) Section 2.3, first line: Please consider space before and after ± in the text
2) Equation (10): Please revise to include “m” in the sigma, just like equation (4)
3) Page 5, line 4: Please revise, Equation 1 -->> Equation ????
4) Page 5, two equations between equations (11) and (12): Please number all equations
5) Figure 2: Please discuss these three figures, and do not leave them without adequate discussion
6) Figures 5-8: Please discuss these figures in depth! These four figures are informative and include the majority of your findings, so please discuss them and add more comments
7) Please add the time unit, e.g., s, ms, etc., to the horizontal axis of plots in Figure 2
8) Captions of Figures 3 and 4: The first letter is Bold; please be consistency
9) Figure 9 is blurry! Please replace it
10) Figure 10: Lack of adequate discussion
11) Figure 11: Please use the same symbols used in the previous plots
12) Figure 11: Please place the (a) and (b) below the figures and inside the plots
In summary, following the results and many efforts to complete this work was exciting. This work could be considered for publication after fulfilling the abovementioned comments if Aerospace seeks an informative article.
Reviewer 3 Report
Summary: The study investigates the phenomenon of mode switching in compressible flow through a rectangular shallow cavity at a Mach number of 0.64. The authors conducted experiments with six cavity models, varying the ratio between the length and depth (l/h) and the ratio between the length and width (l/w) of the cavity. The spectral analysis, wavelet analysis, and correlation analysis are employed to identify the dominant Rossiter-Heller (R-H) modes and quantify mode switching between these modes. The study's main contributions include insights into the effects of cavity geometry on mode switching and the dominance of the second R-H mode in the energy distribution.
Suggestions for improvements:
1. The study identifies the second Rossiter-Heller (R-H) mode as the dominant mode in the energy distribution. However, the manuscript does not provide a detailed explanation of why the second R-H mode is dominant over the other modes. Could the authors elaborate on the underlying mechanisms or flow characteristics that lead to the dominance of the second R-H mode in the investigated cavity flow configurations?
2. The manuscript mentions that mode switching is more significant for two-dimensional and shallower cavities, particularly between the second and third modes. Could the authors provide further insights into the physical phenomena or flow interactions that contribute to the increased significance of mode switching in these specific cavity geometries?
3. The study uses wavelet analysis to determine the strength of variations as a function of both frequency and time. While the results of the wavelet analysis are presented, the manuscript does not discuss the advantages or limitations of using wavelet analysis in the context of analyzing mode switching in cavity flow. Could the authors discuss the rationale behind choosing wavelet analysis and its benefits or limitations in capturing the temporal variations of R-H modes?
Round 2
Reviewer 1 Report
All my objections have been taken into account. In my opinion, the article gained a lot from the development of the analysis.